# Bacterial degradation of ctenophore *Mnemiopsis leidyi* organic matter

Eduard Fadeev,[1] Jennifer H. Hennenfeind,[1] Chie Amano,[1] Zihao Zhao,[1] Katja Klun,[2] Gerhard J. Herndl,[1,3,4] Tinkara Tinta[1,2]

**ABSTRACT** Blooms of gelatinous zooplankton, an important source of protein-rich biomass in coastal waters, often collapse rapidly, releasing large amounts of labile detrital organic matter (OM) into the surrounding water. Although these blooms have the potential to cause major perturbations in the marine ecosystem, their effects on the microbial community and hence on the biogeochemical cycles have yet to be elucidated. We conducted microcosm experiments simulating the scenario experienced by coastal bacterial communities after the decay of a ctenophore (*Mnemiopsis leidyi*) bloom in the northern Adriatic Sea. Within 24 h, a rapid response of bacterial communities to the *M. leidyi* OM was observed, characterized by elevated bacterial biomass production and respiration rates. However, compared to our previous microcosm study of jellyfish (*Aurelia aurita s.l.*), *M. leidyi* OM degradation was characterized by significantly lower bacterial growth efficiency, meaning that the carbon stored in the OM was mostly respired. Combined metagenomic and metaproteomic analysis indicated that the degradation activity was mainly performed by *Pseudoalteromonas*, producing a large amount of proteolytic extracellular enzymes and exhibiting high metabolic activity. Interestingly, the reconstructed metagenome-assembled genome (MAG) of *Pseudoalteromonas phenolica* was almost identical (average nucleotide identity >99%) to the MAG previously reconstructed in our *A. aurita* microcosm study, despite the fundamental genetic and biochemical differences of the two gelatinous zooplankton species. Taken together, our data suggest that blooms of different gelatinous zooplankton are likely triggering a consistent response from natural bacterial communities, with specific bacterial lineages driving the remineralization of the gelatinous OM.

**IMPORTANCE** Jellyfish blooms are increasingly becoming a recurring seasonal event in marine ecosystems, characterized by a rapid build-up of gelatinous biomass that collapses rapidly. Although these blooms have the potential to cause major perturbations, their impact on marine microbial communities is largely unknown. We conducted an incubation experiment simulating a bloom of the ctenophore *Mnemiopsis leidyi* in the Northern Adriatic, where we investigated the bacterial response to the gelatinous biomass. We found that the bacterial communities actively degraded the gelatinous organic matter, and overall showed a striking similarity to the dynamics previously observed after a simulated bloom of the jellyfish *Aurelia aurita s.l.* In both cases, we found that a single bacterial species, *Pseudoalteromonas phenolica*, was responsible for most of the degradation activity. This suggests that blooms of different jellyfish are likely to trigger a consistent response from natural bacterial communities, with specific bacterial species driving the remineralization of gelatinous biomass.

**KEYWORDS** jellyfish, proteases, bacterioplankton, ocean biogeochemistry

B looms of gelatinous zooplankton are increasingly recognized as an important source of organic matter for marine ecosystems worldwide, with possible links to

Address correspondence to Eduard Fadeev, eduard.fadeev@univie.ac.at.

The authors declare no conflict of interest.

See the funding table on p. 19.

anthropogenic pressures and climate change (1). Gelatinous organisms have existed for >600 million years and it is speculated that we may be facing a gelatinous future due to the adaptability of these ancient organisms (2). Particularly, coastal environments are more frequently experiencing recurring seasonal bloom events of native or invasive gelatinous zooplankton species (i.e., the cnidarian subphylum Medusozoa and the phylum Ctenophora, hereinafter collectively coined jellyfish)(3, 4). Jellyfish blooms are characterized by a rapid buildup of biomass up to ~6 mg C m$^{-3}$, and sometimes, particularly in coastal areas, even exceeding ~10 g C m$^{-3}$ (4, 5). These blooms are often short-lived (weeks to months) followed by a sudden collapse of the population, leading to a massive release of detrital organic material (OM) into the ecosystem (6). The released detrital OM has a high protein content and is typically characterized by low carbon (C) to nitrogen (N) molar ratios (C:$N$ =~ 4.5:1), in contrast to the carbon-rich OM released by phytoplankton (C:$N$ = 6.6:1 (7)). About half of the released jelly-OM is in the dissolved phase (<0.8 μm) and, thus, is exclusively accessible to marine pelagic bacteria that respond to it within hours (8). The particulate fraction of the jelly-OM not remineralized by bacteria in the water column sinks to the bottom, where it is degraded mainly by benthic bacterial communities (9 and the reference therein). As marine bacteria are the key drivers of oceanic biogeochemistry (10), unraveling their role in the remineralization of gelatinous OM is important for advancing our understanding of the impact of jellyfish blooms on coastal marine environments and to properly incorporate bacteria-jellyfish interactions into oceanic carbon budgets and biogeochemical cycles (9).

A rapid increase in OM availability during phytoplankton blooms has been shown to trigger a network of metabolic processes within marine bacterial communities (11, 12). However, due to the temporal and spatial patchiness of jellyfish blooms, *in situ* observations of bacteria-jelly OM interactions are scarce and most of the existing knowledge is derived from incubation experiments. Previous studies have shown that bacterial communities thrive on jelly-OM, reaching considerably higher growth rates than otherwise reported for pelagic marine bacteria (9, 13, 14). Our previous microcosm experiments focused on microbial degradation of jelly-OM from the cosmopolitan bloom-forming scyphozoan jellyfish *Aurelia aurita s.l.* revealed that natural marine bacterial communities are capable of consuming most of their detrital OM within 1.5 days (8). We found that, during this short time period, the bacterial community incorporated a large fraction of the introduced organic carbon into its biomass (indicated by a bacterial growth efficiency of 65% ± 27%). Furthermore, the bacterial communities consumed more than 97% of the dissolved proteins and 70% of the dissolved amino acids of the initial jelly-OM pool (8).

The bioavailability of gelatinous detrital OM favors specific bacterial lineages and results in compositional changes in the communities (14–16). During *A. aurita* OM degradation, the taxonomic lineages *Pseudoalteromonas*, *Alteromonas*, and *Vibrio* (all within Gammaproteobacteria) were the most metabolically active members of the bacterial community (17). Using a proteomics approach, we found that *Pseudoalteromonas* played an important role in the initial extracellular degradation of the protein-containing OM, *Alteromonas* processed carbohydrates and organophosphorus compounds, and *Vibrio* exhibited a cheater lifestyle exploiting the by-/end-products of the metabolic processes carried out by the other two lineages (17). Collectively, these observations suggest that the introduction of gelatinous OM can affect natural marine bacterial communities by stimulating the growth of specific bacterial taxa, as well as promoting specific degradation pathways and complex metabolic interactions within the communities.

The term "jellyfish" is not restricted to "true jellyfish" (i.e., class Scyphozoa) and often refers to other gelatinous zooplankton, such as comb jellies (phylum Ctenophora), which also form massive blooms. Some jellyfish and ctenophore species are also invasive and ever since the early 1980s, when the lobed ctenophore *Mnemiopsis leidyi*, native to the western Atlantic Ocean, invaded the Black Sea, its blooms have become an increasingly common phenomenon in Eurasian seas (18, 19). Since 2016, large-scale blooms of *M.*

*leidyi* have recurred annually from mid-summer to late autumn in the northern Adriatic, specifically in the Gulf of Trieste (20). Few previous studies on the interaction between pelagic bacterial communities and *M. leidyi* blooms have shown that similar to other jellyfish, ctenophore detrital OM stimulates bacterial growth and causes a shift in the bacterial communities toward the dominance of Gammaproteobacteria (21). Despite these similarities, differences in life-history traits, genetic background, and hence distinct physiological and biochemical characteristics (e.g., lack of toxins in ctenophores) suggest that the composition of jellyfish and comb jellyfish detrital OM is not the same. We hypothesized that the differences in the OM might lead to different responses in natural bacterial communities. To test that, we replicated previously conducted microcosm experiments on the degradation of *A. aurita* using *M. leidyi* instead and characterized the bacterial metabolic response to ctenophore detrital OM. The highly similar experimental setup facilitated a direct comparison of our results with previous observations obtained for *A. aurita*, revealing differences in bacterial growth efficiency, and identification of a single bacterial lineage that is likely to be a key player in the degradation of gelatinous OM in coastal marine environments.

## RESULTS

The design of the microcosm experiment aimed at mimicking the ecological scenario experienced by pelagic bacterial communities in the Adriatic Sea during the decay of a *M. leidyi* bloom and the subsequent introduction of ctenophore detrital OM. The experiment consisted of six microcosms, inoculated with the same ambient seawater and hence the same microbial community. Three of the microcosms were treated with ctenophore detrital OM (further termed "Cteno-OM," see Materials and Methods section for details on pre-processing of ctenophores), and the other three without amendment served as a control.

Bacterial communities in the Cteno-OM microcosms reached up to one order of magnitude higher cell abundances than the control treatments (Fig. 1A). Bacterial cell abundance peaked after about 21 h in all microcosms, except in one of the Cteno-OM replicates where bacterial abundances continued to increase for a longer time (58 h). During the exponential phase (9–21 h), growth rates differed between the treatments, reaching 0.16–0.21 $h^{-1}$ in the Cteno-OM microcosms compared to 0.02–0.10 $h^{-1}$ in the control. Based on the changes in dissolved organic carbon (DOC) concentrations and the increase in bacterial biomass (Fig. 2), the estimated total bacterial production during the exponential phase was 7–10 µg C $L^{-1}$ $h^{-1}$ and 0.2–0.8 µg C $L^{-1}$ $h^{-1}$ in the Cteno-OM and control microcosms, respectively. Furthermore, bacterial growth efficiency (BGE), calculated from the increase in the bacterial abundance converted into biomass production and the decrease in the concentration of DOC, reached 18%–27% in the Cteno-OM ($n = 2$) compared to 0%–5% in the control microcosms ($n = 2$). Specific gammaproteobacterial taxa (*Alteromonadales*, *Pseudoalteromonadales*, and *Vibrionales*) reached two to three orders of magnitude higher cell-specific productivity in the Cteno-OM than in the control treatments (see Materials and Methods section for details; Fig. 1B). Interestingly, at the peak of the bacterial abundance, *Pseudoalteromonadales* comprised 20%–75% of the total cell-specific bacterial productivity. Cell-specific bacterial respiration showed overall similar trends as biomass production (Fig. 1C).

The exponential growth of the bacterial communities in the Cteno-OM microcosms was associated with decreasing orthophosphate ($PO_4^{3-}$) concentrations (Fig. 2). The increase in total dissolved nitrogen (TDN) concentration in the Cteno-OM microcosms over time was mainly due to a 20-fold increase in ammonium ($NH_4^+$) from 0.54 ± 0.15 to 10.15 ± 0.15 µmol $L^{-1}$, while concentrations of $NO_3^-$ and $NO_2^-$ remained constant (0.06 ± 0.01 and 7.84 ± 0.01 µmol $L^{-1}$, respectively). The initial concentration of dissolved free amino acids (DFAA) was 100 times higher in the Cteno-OM microcosms than in the control and decreased during the entire incubation period from 1.37 ± 0.08 to 0.40 ± 0.12 µmol $L^{-1}$ (Fig. 2). Concentrations of the dissolved combined amino acids (DCAA) in Cteno-OM microcosms increased during the first 58 h from 0.61 ± 0.06 to 3.57 ± 0.14 µmol $L^{-1}$.

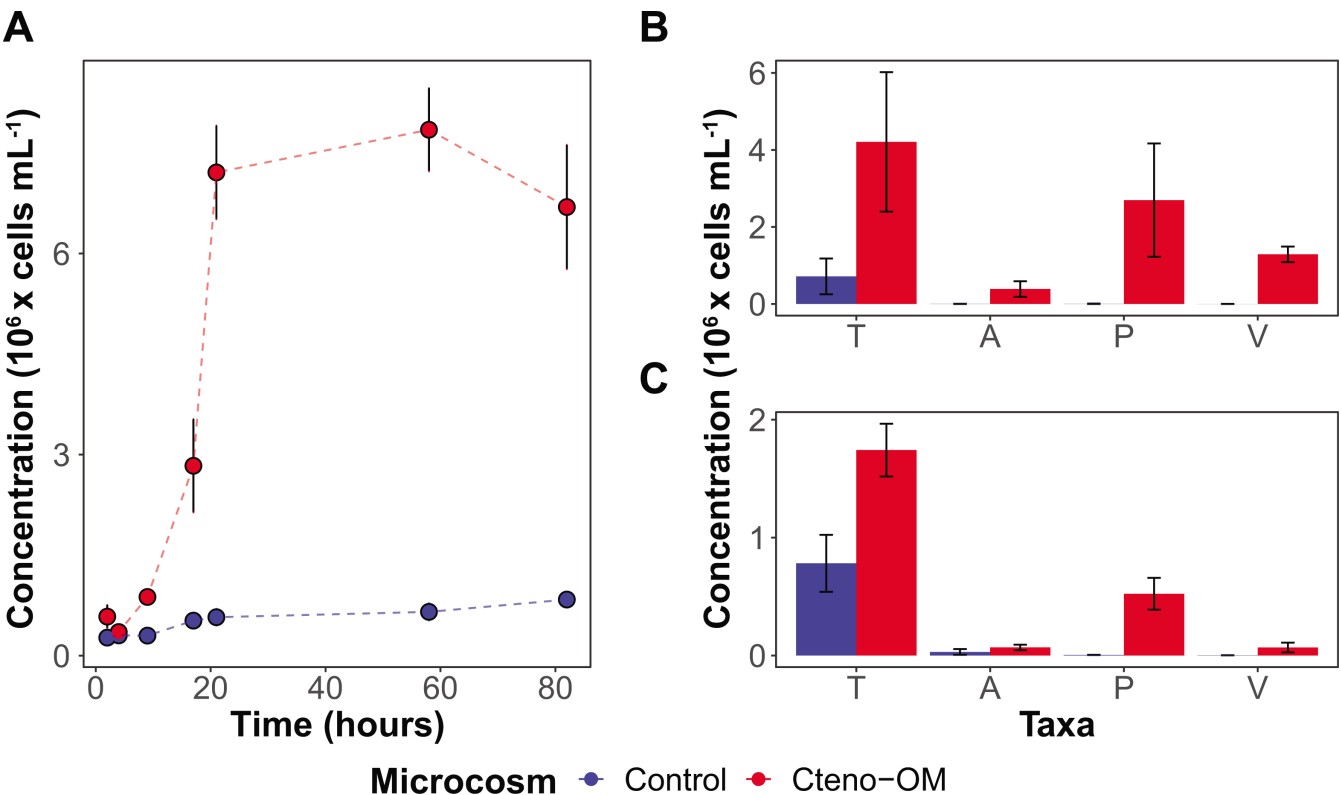

**FIG 1** Abundance and activity of bacterial communities in the microcosms. (A) Total bacterial abundances in Cteno-OM (red) and control (blue) microcosms estimated by DAPI counts. The mean abundance of HPG incorporating (B) and respiring (C) bacterial cells at the peak of the abundance (after 21 h) was estimated using targeted fluorescence in-situ hybridization (FISH) probes. In panels B and C, T—all bacterial cells, A—Alteromonas, P—Pseudoalteromonas, V—Vibrio. Error bars represent standard errors between biological replicates.

Both DFAA and DCAA concentrations remained essentially constant in the control microcosms throughout the experiment (DFAA: from $0.02 \pm 0.01$ to $0.06 \pm 0.01$ µmol $L^{-1}$, DCAA: from $0.73 \pm 0.22$ to $0.70 \pm 0.37$ µmol $L^{-1}$).

Using mass spectrometry, we compared the number of identified *M. leidyi* proteins in the starting Cteno-OM dry material and the Cteno-OM amended microcosms at the peak of bacterial abundance. We found that of the 6,128 *M. leidyi* proteins detected in the Cteno-OM dry material, only half (2,768–4,543) were still detected in the Cteno-OM microcosms at the peak of bacterial abundance (after 21 h). We then analyzed, in each microcosm and the unamended controls, the bacterial proteins associated with cells (further addressed as cellular proteins; >0.22 µm size fraction) and proteins released by the bacteria into the environment (further addressed as extracellular proteins; <0.22 µm size fraction). The cellular fraction of the inoculum contained approximately one-third more bacterial proteins (total of 6,256) than the Cteno-OM (2,170–6,047) and the control (3,088–4,869) microcosms (Fig. S1; Table S1). By contrast, in the extracellular fraction, no major differences were observed in the total number of identified bacterial proteins (inoculum: 3,823, Cteno-OM: 3,310–4,975, control: 3,735–4,026). A total of 461 bacterial proteins were observed across the entire data set (i.e., in both fractions of the Cteno-OM and control mesocosms, as well as in the unamended seawater), 867 proteins were shared between all cellular fractions, and 1,170 proteins were shared between all the extracellular fractions. In each microcosm, half of the bacterial proteins were observed in both size fractions, that is, <0.22 µm and >0.22 µm. However, the composition of proteins was significantly different between the cellular and extracellular fractions (PERMANOVA test; $F_{1,13}=1.73$, $R^2 = 0.11$, $P = 0.02$; Fig. 3), and in both fractions, the composition of proteins was significantly different between the treatments (PERMANOVA test; $F_{2,13}=1.49$, $R^2 = 0.19$, $P = 0.02$).

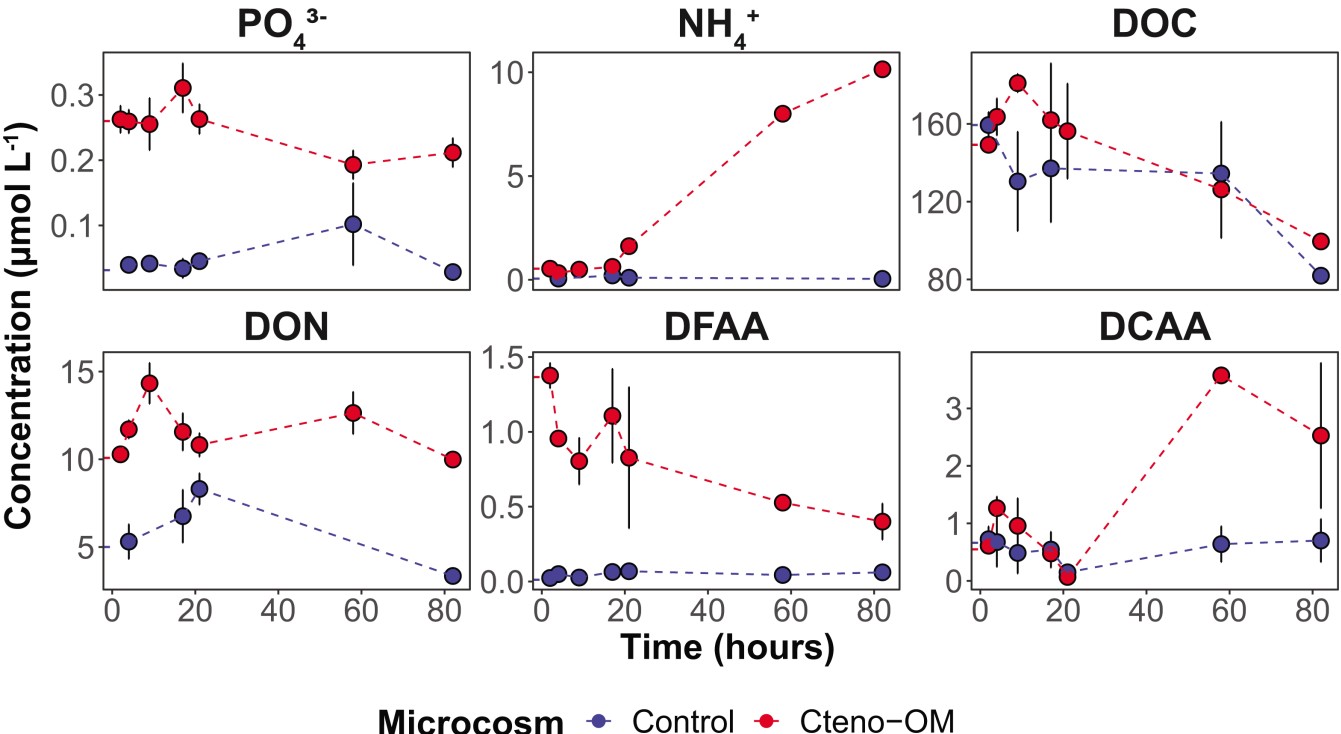

**FIG 2** Nutrient concentrations in the microcosms over time. PO$_4^{3-}$—phosphate, NH$_4^+$—ammonium, DOC—dissolved organic carbon, DON—dissolved organic nitrogen, DFAA—dissolved free amino acids, DCAA—dissolved combined amino acids. Error bars represent standard error between biological replicates. The mean starting concentration in the microcosms is marked along the y-axis.

The taxonomic origin of the identified proteins strongly differed between the bacterial inoculum and the community in the microcosms (Fig. 4). The inoculum community contained mostly proteins from the class Alphaproteobacteria (57% of cellular and 34% of extracellular proteins), followed by Gammaproteobacteria (21% of cellular and 53% of extracellular proteins). In both Cteno-OM and the control microcosms, however, most of the proteins originated from lineages in the Gammaproteobacteria (71%–80% of cellular and 63%–71% of extracellular proteins), while Alphaproteobacteria comprised only 16%–21% of cellular and 23%–26% of extracellular proteins at the peak of bacterial abundance. The order *Alteromonadales* (class Gammaproteobacteria) comprised 52%–63% of all cellular proteins in the Cteno-OM microcosms and 28%–43% of all cellular proteins in the control. Less pronounced differences were observed in the proportional abundance of *Alteromonadales*-associated proteins in the extracellular fraction, where they comprised 28%–37% of all extracellular proteins in the Cteno-OM microcosms and 22%–34% of all extracellular proteins in the control. In the control microcosms, the order *Oceanospirillales* (class Gammaproteobacteria) exhibited a higher relative abundance of proteins than in the Cteno-OM treatment (16%–23% vs 9%–12% of the proteins in the cellular fraction, 28%–37% and 22%–34% of the proteins in the extracellular fraction). The order *Pelagibacterales*, comprising the largest portion of proteins among Alphaproteobacteria did not show differences in the relative abundance of proteins between the two treatments, comprising 4%–26% of the proteins in the cellular fraction and 8%–25% of the proteins in the extracellular fraction.

To identify significantly enriched proteins between the Cteno-OM and the control microcosms, we implemented the DESeq2 enrichment algorithm on both the cellular and the extracellular protein data set (Fig. 5). We identified 490 and 603 proteins in the cellular and extracellular fractions, respectively, which were significantly enriched in the Cteno-OM microcosms (adjusted *P* value < 0.1; Table S2). In both fractions combined, the bacterial genus with the largest number of enriched proteins was *Pseudoalteromonas* (365 proteins; Fig. 6). The enriched proteins of this genus were mostly associated with the

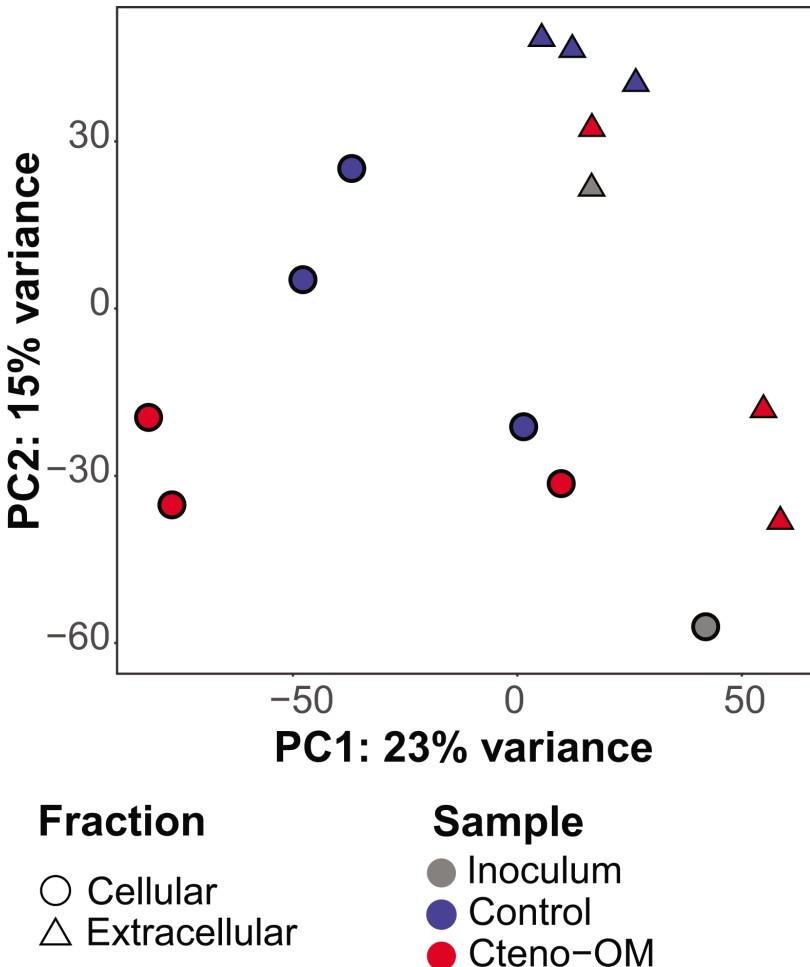

**FIG 3** Dissimilarity of the protein composition in the microcosms and the inoculum bacterial communities. Principal component analysis (PCA) was performed on variance-stabilized protein composition in both the cellular (circle) and extracellular (triangle) fractions. Except for the inoculum that had no replicates, the dots represent biological replicates in each fraction.

Clusters of Orthologous Genes (COG) categories "J"—Translation, ribosomal structure, and biogenesis (mostly various tRNA synthetases and other ribosomal proteins), "E"—Amino acid transport and metabolism (among them Leucyl-, Xaa-Pro-, and N-aminopeptidases), and "O"—Posttranslational modification and protein turnover (among them Lon and Serine proteases, as well as various peptidases). The genus *Alteromonas* also contained a large number of significantly enriched proteins (48 proteins; Fig. 6), associated with COG categories "O"—Posttranslational modification and protein turnover (7 out of the total eight proteins were ATP-dependent Lon protease) and "J"—Translation, ribosomal structure, and biogenesis (containing various tRNA synthetases and other ribosomal proteins). The second largest group of significantly enriched proteins in the Cteno-OM microcosms was affiliated to the genus *Vibrio* (61 proteins; Fig. 6) with the enriched proteins associated with COG categories "G"—Carbohydrate transport and metabolism (proteins of various metabolic functions), "J"—Translation, ribosomal structure, and biogenesis (mostly ribosomal proteins), and "E"—Amino acid transport and metabolism (among which various proteins associated with ABC-type transport systems). In the control microcosms, we found 302 and 339 significantly enriched proteins in the cellular and extracellular fractions, respectively (Table S2). In both fractions, the enriched proteins were mostly associated with various genera within the orders *Alteromonadales* and *Pelagibacterales*, with no clear trends in their COG affiliations.

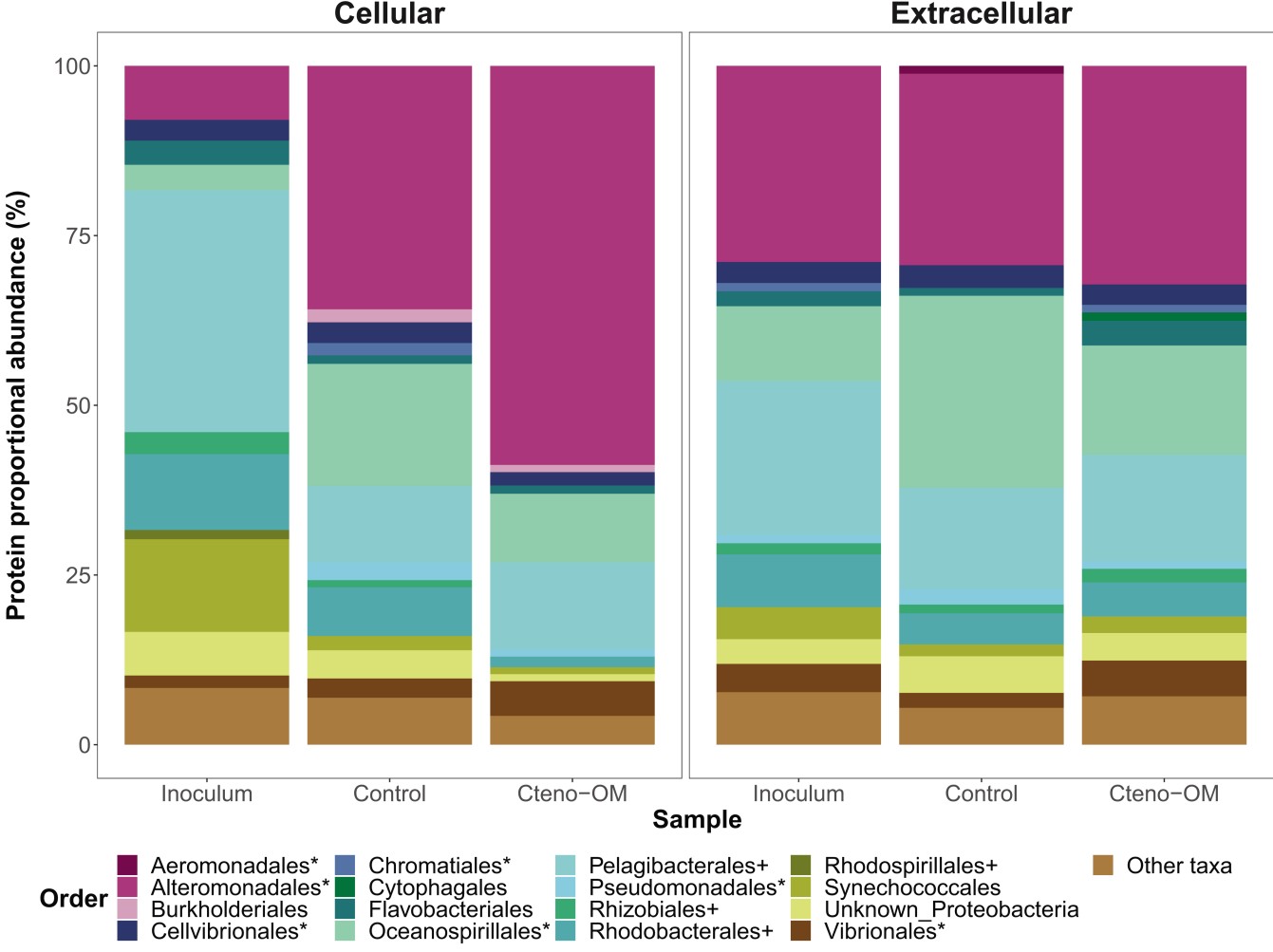

**FIG 4** Taxonomic breakdown of bacterial proteins detected in the cellular and extracellular fractions at the peak of bacterial abundance (after 21 h). The stack bar for the two treatments (Cteno-OM and unamended control) represents the mean value of three replicates. Orders within Alphaproteobacteria and Gammaproteobacteria are indicated in the legend with a (+) and a (*) respectively.

Using a co-assembled metagenome of DNA libraries from both the Cteno-OM and the control microcosms as well as the inoculum seawater sample (described in Materials and Methods), we reconstructed 33 bacterial metagenome-assembled genomes (MAGs), all larger than 1.2 Mbp, with estimated completeness of >75%, and redundancy <8% (Fig. 7). The coverage of the MAGs by reads from each library (i.e., different samples) revealed that the DNA of two MAGs was notably more abundant in the Cteno-OM than in the control treatment. These two MAGs were taxonomically affiliated with the bacterial species *Marinobacterium jannaschii* and *Pseudoalteromonas phenolica* (Fig. S2). The reconstructed *P. phenolica* MAG (Bin_84) had the highest read coverage among all MAGs and comprised 506 contigs with a total of 4,145 coding genes (overall length 4.2 Mbp) and was highly similar to the previously reconstructed MAG in the *A. aurita s.l.* degradation experiment [average nucleotide identity—99%; (17)]. The reconstructed *P. phenolica* MAG shared 1,143 genes with all other *P. phenolica* genomes (i.e., core genes) and contained 122 genes not present in any other genome (Fig. S3; Table S3).

Of all the MAGs reconstructed, the *P. phenolica* MAG was associated with the largest number of proteins in the microcosms (194–505 in the Cteno-OM and 209–287 in the control treatments). In the Cteno-OM treatments, these proteins accounted for 21%–27% in the cellular and 7%–8% in the extracellular fraction of all proteins in the data set (Fig. 8). To characterize the potentially enriched metabolism of *P. phenolica* in the Cteno-OM

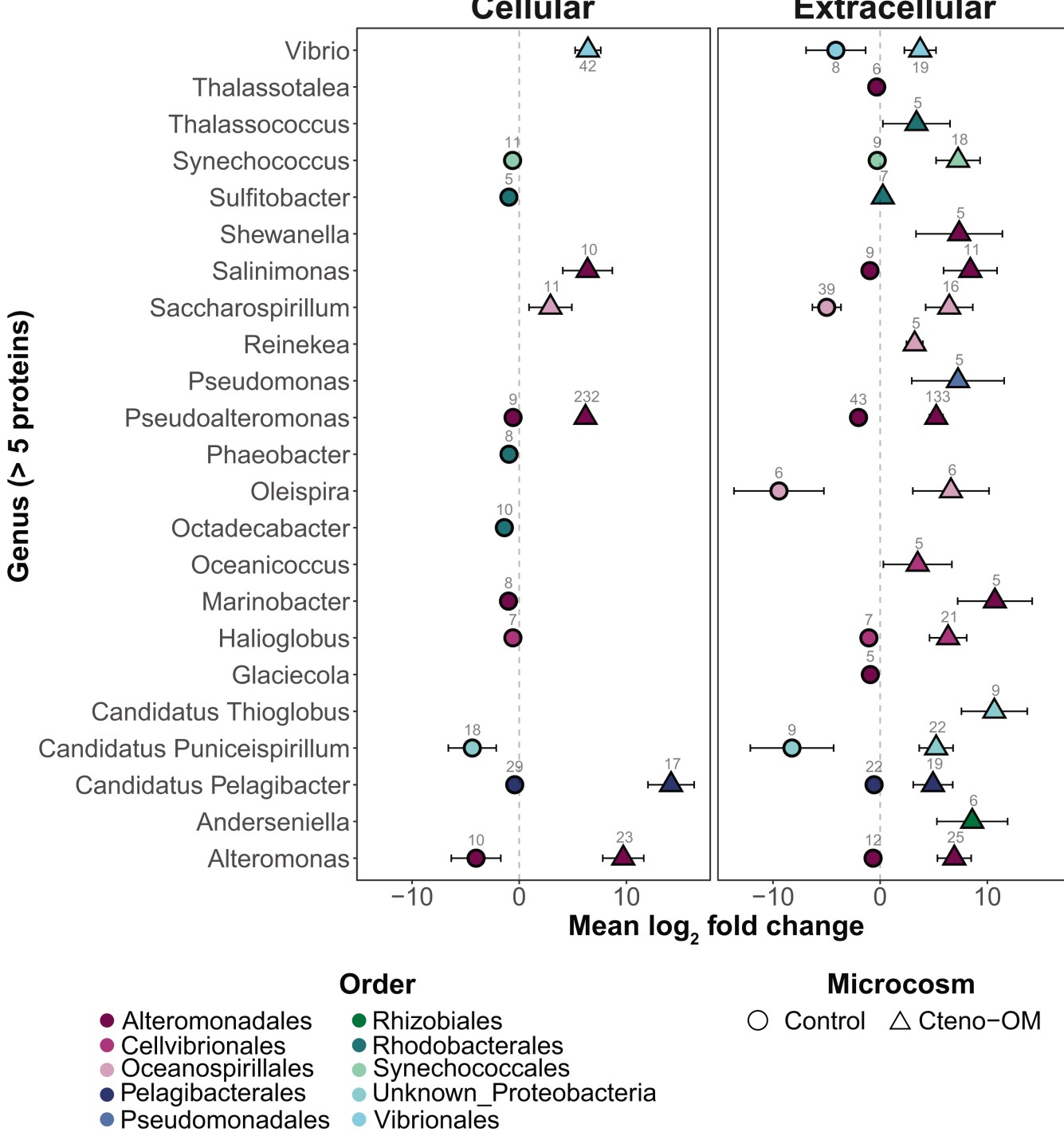

**FIG 5** Differential abundance of proteins according to taxonomic origins between Cteno-OM and control microcosms. The mean $\log_2$ fold-change of significantly enriched proteins was calculated according to a grouping of distinct genera and fractions (cellular and extracellular). Positive $\log_2$ fold-change values represent enrichment in Cteno-OM microcosms, and negative values represent enrichment in control microcosms.

treatments, we reconstructed its metabolic pathways using the genomic information in the MAG and combined it with the identified significantly enriched proteins. Among those *P. phenolica* MAG-associated proteins, 149 proteins were significantly enriched in the Cteno-OM treatments (in contrast to only 16 significantly enriched proteins in the control treatments; Table S3). The genomic information of the *P. phenolica* MAG contained numerous complete metabolic pathways (completeness defined by the presence

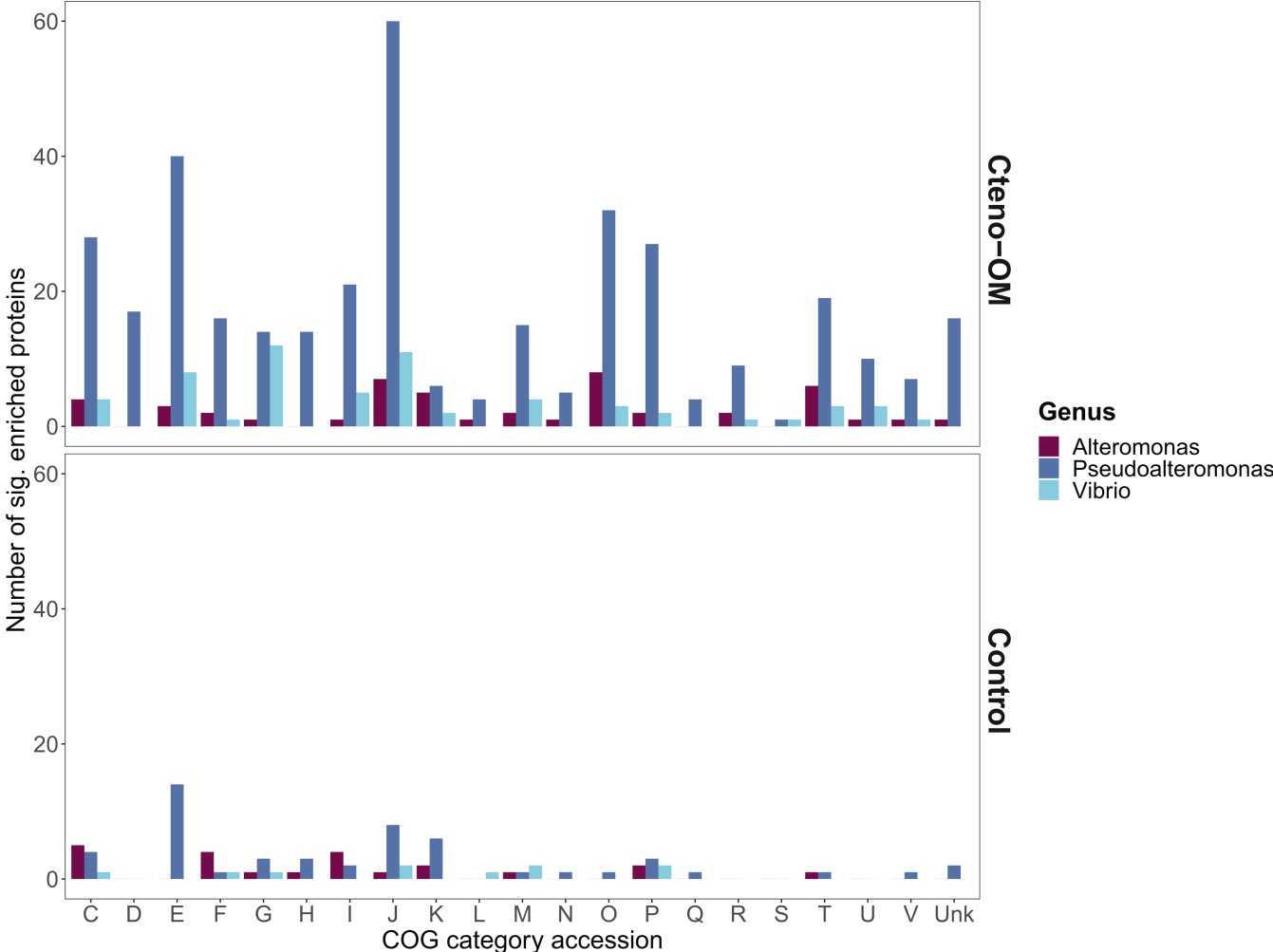

**FIG 6** Overview of significantly enriched proteins in the most dominant Gammaproteobacteria genera according to a cluster of ortholog categories. "C"— Energy production and conversion, "D"—Cell cycle control, cell division, and chromosome partitioning, "E"—Amino acid transport and metabolism, "F"— Nucleotide transport and metabolism, "G"—Carbohydrate transport and metabolism, "H"—Coenzyme transport and metabolism, "I"—Lipid transport and metabolism, "J"—Translation, ribosomal structure, and biogenesis, "K"—Transcription, "L"—Replication, recombination, and repair, "M"—Cell wall/membrane/envelope biogenesis, "N"—Cell motility, "O"—Posttranslational modification, protein turnover, and chaperones, "P"—Inorganic ion transport and metabolism, "Q"—Secondary metabolites biosynthesis, transport, and catabolism, "R"—General function prediction only, "S"—Function unknown, "T"—Signal transduction mechanisms, "U"—Intracellular trafficking, secretion, and vesicular transport, "V"—Defense mechanisms, and "Unk"—unclassified.

of >70% of the required genes; Table 1). We found significantly enriched proteins associated with glycolysis (K01803—EC:5.3.1.1) and gluconeogenesis (K01610—EC:4.1.1.49) pathways, as well as with pyruvate oxidation (K00382—EC:1.8.1.4). Among the enriched proteins, we also found two subunits (alpha—K02111 and beta—K02112) of F-type ATPase (EC:7.1.2.2) and one subunit (iron-sulfur subunit—K00411) of the Cytochrome c reductase (EC:7.1.1.8), both associated with oxidative phosphorylation. These were complemented by the enrichment of two different proteins (K03183—EC:2.1.1.201 and K00568—EC:2.1.1.222) in the ubiquinone biosynthesis pathway, a key redox cofactor of the electron transfer chain. In addition, we found enrichment of the acyl-CoA dehydrogenase (K00249—EC:1.3.8.7) playing a key role in beta-oxidation of fatty acids. However, this enzyme is also involved in the leucine degradation pathway, in which we also observed an enrichment of leucine dehydrogenase (K00263—EC:1.4.1.9) and dihydrolipoamide dehydrogenase (K00382—EC:1.8.1.4). Also, enriched proteins such as glutamate dehydrogenase (K00260—EC:1.4.1.2) are potentially linked to the degradation of other amino acids, such as alanine, arginine, and of taurine. We further observed

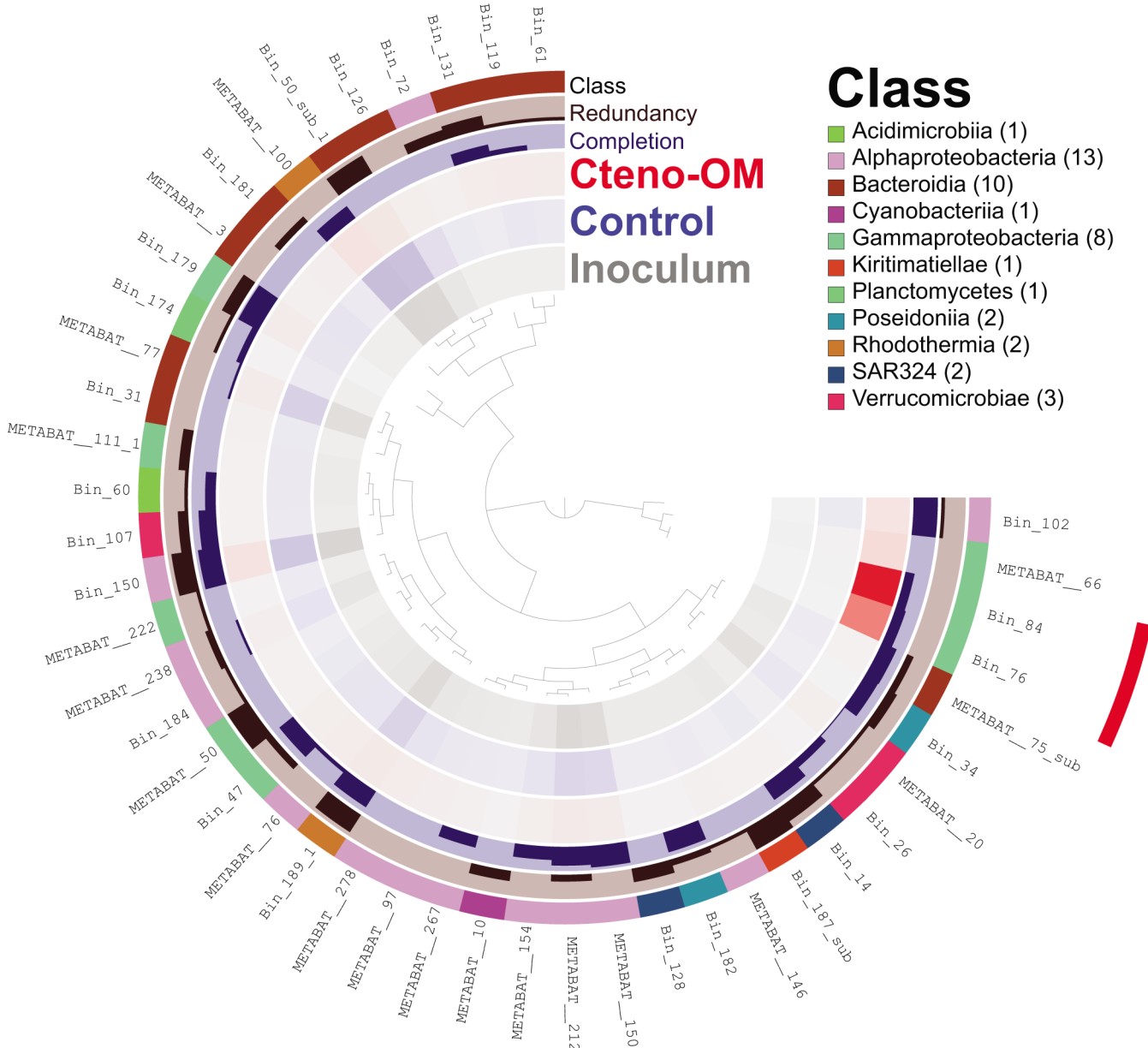

**FIG 7** Reconstructed MAGs from the co-assembled metagenome. The color density in the three inner rings represents the normalized average coverage of each MAG by reads from each metagenome (Cteno-OM, control, and inoculum). The MAGs were clustered using Euclidian distance between their read coverages. The MAGs marked in red had higher read coverage in the Cteno-OM metagenome. Completion represents the estimated completeness of the MAG ranging from 75% to 100%. The redundancy represents the estimated sequence contamination in the MAG ranging from 0% to 8%.

the enrichment of various proteins associated with protein metabolism, such as the protein export system SecY/Sec61 (K03070—EC:7.4.2.8) and signal peptidases (K03100—EC:3.4.21.89).

## DISCUSSION

Gelatinous zooplankton include a group of diverse organisms that share several traits such as planktonic lifestyle and trophic position. These organisms are commonly referred to as "jellyfish" and are often lumped together in biogeochemical oceanic models and budgets. However, despite their similarities, these organisms are characterized by different life-history traits and are genetically, and therefore, physiologically and biochemically distinct. To compare the biochemical composition of the ctenophore

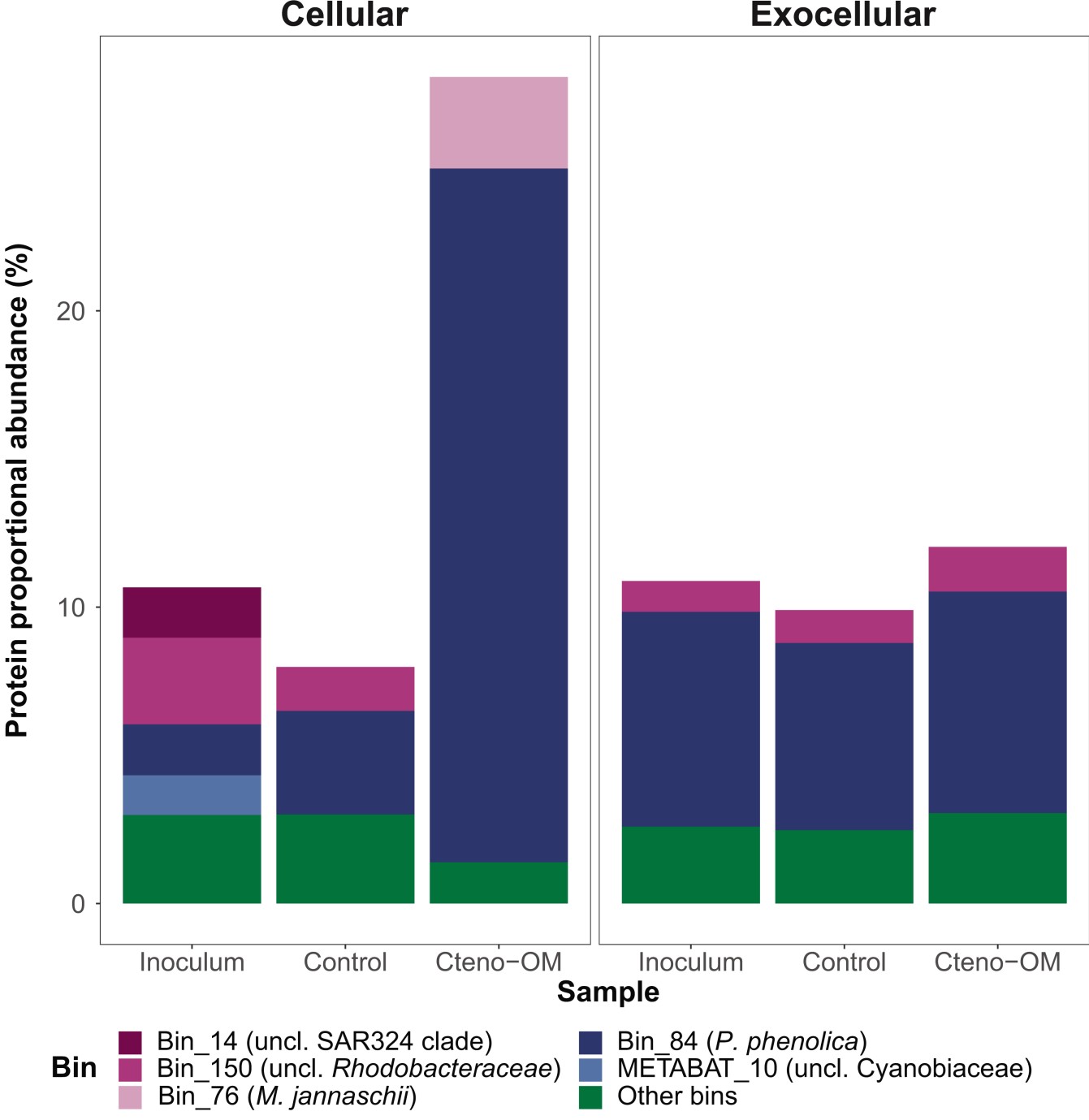

**FIG 8** Proportional abundance of proteins associated with MAGs. The MAGs with a total protein proportion below 1% were grouped under "other bins."

*M. leidyi* OM with that of the scyphozoan jellyfish *A. aurita s.l.*, we conducted leaching experiments in which we dissolved gelatinous OM in artificial seawater without the presence of bacterial communities and measured dissolved organic and inorganic nutrients released over 24 h. We found that the *A. aurita s.l.* contains more than twice the amount of dissolved organic carbon, total dissolved nitrogen, and phosphate (Table 2). This raised the question of whether the biochemical differences affect the "fate" of the gelatinous detrital OM. Do marine ambient bacterial communities exhibit the same response to gelatinous OM regardless of its source and distinct biochemical features?

**TABLE 1** Reconstructed metabolic pathways from the *P. phenolica* MAG[a]

| KEGG subcategory | Pathway (KEGG module) | Completeness | KO hits in the MAG |
|---|---|---|---|
| Arginine and proline metabolism | Ornithine biosynthesis, glutamate => ornithine (M00028) | 100% | K01438, K00145, K00821, K00930, K14682 |
| | Arginine biosynthesis, glutamate => acetylcitrulline => arginine (M00845) | 71% | K01438, K00145, K00821, K01940, K01755 |
| ATP synthesis | Cytochrome bc1 complex respiratory unit (M00151) | 100% | K00413, K00412, K00411 |
| | F-type ATPase, prokaryotes, and chloroplasts (M00157) | 100% | K02109, K02108, K02111, K02110, K02112, K02113, K02114, K02115 |
| Carbon fixation | Reductive pentose phosphate cycle, ribulose-5*P* => glyceraldehyde-3P (M00166) | 75% | K00134, K00927, K00855 |
| Central carbohydrate metabolism | Pyruvate oxidation, pyruvate => acetyl-CoA (M00307) | 100% | K00163, K00382, K00627 |
| | PRPP biosynthesis, ribose 5*P* => PRPP (M00005) | 100% | K00948 |
| | Gluconeogenesis, oxaloacetate => fructose-6P (M00003) | 88% | K00134, K01624, K00927, K01803, K01689, K15633, K01610 |
| | Glycolysis, core module involving three-carbon compounds (M00002) | 83% | K00134, K00927, K01803, K01689, K15633 |
| | Citrate cycle, second carbon oxidation, 2-oxoglutarate => oxaloacetate (M00011) | 80% | K01676, K01679, K00382, K00242, K00240, K00241, K01902, K01903, K00164, K00239, K00658 |
| | Citrate cycle (TCA cycle, Krebs cycle) (M00009) | 75% | K01676, K01679, K00382, K01682, K00242, K00240, K00241, K01902, K01903, K01647, K00164, K00239, K00658 |
| | Pentose phosphate pathway, non-oxidative phase, fructose 6*P* => ribose 5P (M00007) | 75% | K00616, K01783, K01807 |
| Cofactor and vitamin metabolism | Siroheme biosynthesis, glutamyl-tRNA => siroheme (M00846) | 92% | K02496, K01749, K01719, K01698, K01845, K02492, K02303 |
| | Heme biosynthesis, plants and bacteria, glutamate => heme (M00121) | 90% | K01885, K01749, K01719, K01698, K01599, K02495, K01845, K02492, K00228, K01772 |
| | Ubiquinone biosynthesis, prokaryotes, chorismate (+ polyprenyl-PP) =>ubiquinol (M00117) | 89% | K03179, K03181, K00568, K03184, K03186, K03183, K03185, K03182 |
| | Pimeloyl-ACP biosynthesis, BioC-BioH pathway, malonyl-ACP => pimeloyl-ACP (M00572) | 83% | K00059, K02372, K02170, K02169, K09458 |
| | Heme biosynthesis, bacteria, glutamyl-tRNA => coproporphyrin III =>heme (M00926) | 78% | K01749, K01719, K01698, K01599, K01845, K02492, K01772 |
| | Heme biosynthesis, animals and fungi, glycine => heme (M00868) | 75% | K01749, K01719, K01698, K01599, K00228, K01772 |
| Fatty acid metabolism | Fatty acid biosynthesis, elongation (M00083) | 100% | K00209, K00059, K01716, K02372, K09458 |
| | Beta-oxidation (M00087) | 100% | K01825, K00632, K01782, K01692, K00249, K06445 |
| Histidine metabolism | Histidine degradation, histidine => N-formiminoglutamate =>glutamate (M00045) | 100% | K01479, K01468, K01712, K01745 |

(*Continued on next page*)

**TABLE 1** Reconstructed metabolic pathways from the *P. phenolica* MAG[a] (*Continued*)

| KEGG subcategory | Pathway (KEGG module) | Completeness | KO hits in the MAG |
|---|---|---|---|
| Lysine metabolism | Lysine biosynthesis, succinyl-DAP pathway, aspartate => lysine (M00016) | 89% | K00133, K00674, K01586, K01778, <u>K00821</u>, K00215, K01714, K01439 |
| Other carbohydrate metabolism | Glyoxylate cycle (M00012) | 80% | K01638, <u>K01637</u>, K01682, <u>K01647</u> |
| Pyrimidine metabolism | Polyamine biosynthesis, arginine => agmatine => putrescine => spermidine (M00133) | 75% | K01611, K00797, <u>K01480</u> |
| | Uridine monophosphate biosynthesis, glutamine (+ PRPP) =>UMP (M00051) | 83% | K01591, K00762, <u>K01955</u>, K01956, K01465, K00254 |

[a]For each pathway, the list of KO definitions represents genes identified in the MAG, and the underlined definitions were further identified as significantly enriched proteins.

Understanding this is crucial for correctly incorporating bacteria-jellyfish interactions into oceanic biogeochemical models.

To start answering these questions, we replicated the experimental design of our previously published study on bacterial degradation of *A. aurita s.l.* OM (8), to investigate the response of pelagic bacterial communities to OM of *M. leidyi*. The addition of ctenophore OM to the microcosms promoted rapid bacterial growth and activity, similar to the *A. aurita* treatments (Table 2). However, the biomass reached by the bacterial communities in Cteno-OM microcosms was much lower compared to the *A. aurita s.l.* experiment, likely due to the higher DOC content in the latter (ca. 200 µmol L$^{-1}$ of *M. leidyi* OM and ca. 400 µmol L$^{-1}$ of *A. aurita* OM; Table 2). Bacterial growth in Cteno-OM microcosms, supported by DOC and phosphate availability, led to rapid processing of the Cteno-OM and accumulation of ammonia and dissolved combined amino acids. The DCAA accumulation most likely represented an enhanced extracellular enzymatic cleavage of proteins and peptides, as well as slower bacterial utilization rates compared to DFAA (22, 23). Additional evidence for protein degradation was the much lower number of *M. leidyi* proteins identified in the Cteno-OM microcosms at the peak of bacterial abundance as compared to the initial *M. leidyi* OM protein content.

**TABLE 2** Comparison between *M. leidyi* and *A. aurita s.l.* leaching and microcosm experiments[a]

| Biochemical characteristics of OM powder | | |
|---|---|---|
| | *M. leidyi* (n = 2) | *A. auritas.l.* (n = 4) |
| DOC | 227–228 | 397–473 |
| TDN | 60–70 | 119–137 |
| NH$_4^+$ | 5–10 | 3–16 |
| DIN | 5–10 | 0.3–23 |
| DON | 50–65 | 112–128 |
| PO$_4^{3-}$ | 2 | 6–7 |
| C:N[b] | 3.2–3.8 | 3.3–3.4 |
| **Bacterial community growth parameters** | | |
| | *M. leidyi* (n = 2) | *A. aurita* (n = 2) |
| BCD (µg C L$^{-1}$ h$^{-1}$) | 46–60 | 21–45 |
| BB (µg C L$^{-1}$) | 144–169 | 175–216 |
| µ (h$^{-1}$) | 0.07–0.08 | 0.08 |
| BP (µg C L$^{-1}$ h$^{-1}$) | 11–13 | 14–17 |
| BR (µg C L$^{-1}$ h$^{-1}$) | 33–49 | 3–31 |
| BGE (%) | 18–27 | 31–83 |

[a]All biochemical values represent concentrations of dissolved analyte (<0.8 µm) in µmol g$^{-1}$ powder (i.e., per dry weight of a subsample of *M. leidyi* or *A. aurita s.l.* representative populations). Due to a substantially different community dynamic, one replicate in each experiment was excluded from growth estimates. DOC—dissolved organic carbon; TDN—total dissolved nitrogen; DIN—dissolved inorganic nitrogen = NH$_4^+$+NO$_2^-$+NO$_3^-$; DON—dissolved organic nitrogen = TDN-DIN; BCD—bacterial carbon demand; BB —bacterial biomass; µ—growth rate; BP—bacterial production; BR—bacterial respiration; BGE—bacterial growth efficiency.
[b]Note that C:N ratio is calculated for the dissolved fraction, that is, DOC:DON.

The introduction of *M. leidyi* detrital OM led to increased activity of *Vibrionales*, *Pseudoalteromonadales*, and *Alteromonadales*, the latter is often found enriched in natural seawater communities during phytoplankton blooms (11, 12, 24). At the peak of bacterial abundance, the order *Alteromonadales* accounted for more than half of the bacterial proteins identified in the Cteno-OM microcosms. Protein enrichment analysis between the Cteno-OM and the control microcosms revealed that most of the enriched proteins originated from the genera *Pseudoalteromonas* and *Alteromonas*, and were linked to protein hydrolysis, as well as translational and ribosomal processes (t-RNAs synthetases). To elucidate the specific metabolic pathways, these enriched proteins were involved in, we reconstructed a *P. phenolica* MAG, which dominated the bacterial community at the peak of bacterial abundance. The reconstructed *P. phenolica* MAG was genomically identical to a *Pseudoalteromonas* MAG acquired from the *A. aurita s.l.* microcosm experiment (17). This bacterial species has been previously characterized as a major producer of proteases, lipases, and other hydrolytic enzymes (25, 26). Among other enzymes, it also possesses collagenases that play a key role in the microbial recycling of OM during the decay phase of jellyfish blooms (17). In our study, we found that *P. phenolica* was actively involved in the hydrolysis of the protein-rich ctenophore OM through the production of a wide range of proteases. The hydrolysis products (i.e., amino acids) were then taken up by *P. phenolica* cells and further degraded through various amino acid metabolic pathways (e.g., leucine and taurine). We also found evidence of lipid oxidation associated with the ctenophore OM (e.g., acyl-CoA dehydrogenase). The products of these degradation pathways have fueled the central metabolism of *P. phenolica,* thereby elevating its activity, which most likely led to the high abundance of metabolically active *Pseudoalteromonas* as observed by our epifluorescence microscopy-based approaches (Fig. 1B and C).

The *M. leidyi* and *A. aurita s.l.* microcosm experiments used ambient seawater from the same geographic location (same sampling station), but 1 year apart. The experiments showed that the magnitude of the bacterial response to the addition of gelatinous OM varied according to its biochemical composition and concentration of specific compounds. However, the striking similarity between the key bacterial players in both experiments and the consistency with previous jellyfish incubation experiments (8, 15, 21, 27) suggests that copiotrophic bacterial lineages, such as *P. phenolica*, drive a fairly consistent metabolic response of marine bacterial communities to natural blooms of different gelatinous organisms. Therefore, we postulate that to assess the biogeochemical impact of "jellyfish" blooms, special attention must be paid to the identification of bacterial lineages with specific enzymatic and metabolic capabilities for the remineralization of gelatinous OM.

## MATERIALS AND METHODS

### Preparation of *M. leidyi* OM powder and its biochemical characterization

A total of 21 specimens of *M. leidyi* were collected in the Gulf of Trieste, northern Adriatic Sea, during the summer bloom in August 2019. The ctenophores were collected on different days and at different locations to account for potential spatial and temporal heterogeneity in the bloom population. Individuals were sampled from the surface of the water column using a large acid-cleaned plastic bucket and stored in zip-lock bags at −20°C within 1 h. Each ctenophore was then freeze-dried at −45°C for 7 days and the dry material of all ctenophores was pooled and homogenized with a sterilized pestle and agate mortar. The Cteno-OM powder was then stored in acid- and Milli-Q water-rinsed and combusted glass vials at −20°C. To minimize the risk of contamination and degradation of the Cteno-OM powder, care was taken to work under sterile conditions and on ice at all intermediate steps.

To determine its chemical composition, leaching experiments were conducted as follows: 250 mg of the Cteno-OM powder was dissolved in 1 L of artificial seawater (prepared according to (28) in an acid- and Milli Q water-rinsed and combusted glass

Erlenmeyer flask. To ensure maximized dissolution of the powder, the Erlenmeyer flask was placed on a shaker in the dark at room temperature for 24 h. From each flask, a technical duplicate was collected for biochemical characterization as described below. Background values for each parameter were determined by subsampling the artificial seawater and subsequently subtracting the respective value from that measured in the Cteno-OM treatment. In addition, the absence of possible bacterial contamination was verified using microscopy at the end of each experiment.

## Experimental design of the microcosms

In August 2019, 60 L of seawater was collected from a 5 m depth in the center of the Gulf of Trieste and aged in acid-washed and Milli-Q water-rinsed 20 L Nalgene carboys (ThermoFisher Scientific, Rockford, IL, USA) for about 1 month at room temperature in the dark, and then filtered through an 0.22 µm polycarbonate filter to remove microbial cells. In September 2019, an additional water sample was collected at the same sampling location and pre-filtered through a 1.2 µm polycarbonate filter to remove most of the non-bacterial particulate matter from the sample. Then, the aged and the "fresh" seawater were distributed into six 10 L borosilicate glass bottles in a 9:1 ratio. Based on an average of ca. 100 *M. leidyi* specimens m$^{-3}$ observed during sampling and their dry weight of ca. 1 g each, a total of 1 g of Cteno-OM powder was added to three microcosms, to reach a final concentration of 100 mg L$^{-1}$. Three other microcosms, with no Cteno-OM amendment, served as control. All bottles were incubated in the dark at the ambient seawater temperature (ca. 24°C) and mixed gently prior to subsampling.

## Dissolved organic carbon and nitrogen

Samples for dissolved organic carbon (DOC) and total dissolved nitrogen (TDN) were filtered through combusted GF/F filters (Whatman, Maidstone, UK) using an acid-washed, Milli-Q water-rinsed, and combusted glass filtration system. Approximately 30 mL of the filtrate was collected into acid-, Milli-Q water-rinsed, and combusted glass vials and acidified with 12 M HCl (~100 µL per ~20 mL of sample) to reach a final pH <2 and stored at 4°C until analysis. Both DOC and TDN analyses were performed by the high temperature catalytic method using a Shimadzu TOC-L analyser (Shimadzu, Kyoto, Japan) equipped with a total nitrogen unit (29). The calibration for non-purgeable organic carbon was done with potassium phthalate and for TDN potassium nitrate was used. The results were validated with Deep-Sea Reference (DSR) water for DOC and TDN (CRM Program, Hansell Lab). The precision of the method expressed as RSD % was <2%.

## Dissolved inorganic nutrients

Dissolved inorganic nitrogen compounds ($NH_4^+$, $NO_2^-$, and $NO_3^-$) and dissolved inorganic phosphorus ($PO_4^{3-}$) concentrations were determined spectrophotometrically by QuAAtro segmented flow analysis (Seal Analytical, Norderstedt, Germany) following standard methods (30). The validation and accuracy of the results were checked with reference material (Kanso Technos, Osaka, Japan) before and after sample analyses. The quality control is performed annually by participating in an intercalibration program (QUASIMEME Laboratory Performance Study).

## Dissolved amino acid analysis

Samples for total dissolved amino acid analyses were filtered through combusted GF/F filters (Whatman, Maidstone, UK) using an acid-washed, Milli-Q water-rinsed, and combusted glass filtration system. The filtrate was collected in a dark glass vial and stored at −20°C until analysis. For each sample, two technical replicates of approximately 4 mL were collected. Samples were analyzed for dissolved free amino acids (DFAA) and total dissolved hydrolyzable amino acids (TDHAA). For TDHAA analysis, 500 µL of the sample was first hydrolyzed according to reference (31) with some modifications described elsewhere (8). For DFAA analysis, 500 µL of the sample was used directly. For

both measurements, the samples were pipetted into acid-washed, Milli-Q water-rinsed, and combusted glass HPLC ampules and analyzed on a Shimadzu Nexera X2 ultra high-performance liquid chromatograph (UHPLC) with a fluorescence detector (RF-20A XS; Shimadzu, Kyoto, Japan). Pre-column derivatization was applied with *ortho*-phthaldialdehyde (OPA) following (32) with slight modifications (8). The concentration of dissolved combined amino acids (DCAA) was calculated as the difference between TDHAA and DFAA.

## Total bacterial abundance

To determine the total bacterial abundance, two technical replicates of 1.5 mL were fixed with 0.2 µm-filtered 37% formaldehyde (2% final concentration) and immediately stored at −80°C. Then, using a glass filtration system and a vacuum pump at low pressure (<200 mbar), 1 mL of each replicate was filtered onto an 0.22 µm white polycarbonate filter supported by an 0.45 µm cellulose acetate filter, and stained with 2 µg mL$^{-1}$ 4′,6-diamidino-2-phenylindole (DAPI) in Vectashield (Vector Laboratories, Newark, CA, USA). The cells were enumerated using a Zeiss Axio Imager M2 epifluorescence microscope (Carl Zeiss AG, Oberkochen, Germany) at 1,250× magnification and the DAPI filter set (Ex/Em = 358/461 nm). The bacterial abundance was calculated based on the average number of cells from at least 20 counting fields with 20–200 cells enumerated per counting field.

## Bacterial cell-specific respiration and biomass production

The abundance of respiring bacterial cells was determined at a single-cell level using the BacLight Redox Sensor Green Vitality Kit (ThermoFisher Scientific, Rockford, IL, USA). The Redox Sensor Green (RSG) dye produces green fluorescence (Ex/Em = 495/519 nm) when modified by bacterial reductases, many of which are part of electron transport systems and can therefore serve as a proxy for bacterial respiration (33, 34). From each sample, a technical triplicate of 5 mL was spiked with RSG (final concentration of 1 µM), incubated in cultivation tubes with vent caps at *in situ* temperature in the dark for 30 min, fixed with 0.2 µm filtered 37% formaldehyde (2% final concentration) and stored at −80°C. Prior to the microscopic analysis, the samples were filtered and stained with DAPI (see Total Bacterial Abundance section).

The biomass production of the bacterial community was determined at the single-cell level based on the incorporation rates of the methionine analog L-homopropargylglycine (HPG) into newly synthesized bacterial proteins (35). The incorporation of HPG was detected using click-chemistry, where the alkyne-modified HPG is detected with Alexa Fluor 488 azide (Ex/Em = 490/525 nm), following the manufacturer's protocol (Click-iT HPG Alexa Fluor 488 Protein Synthesis Assay Kit; ThermoFisher Scientific, Rockford, IL, USA). From each sample, a technical triplicate of 5 mL was spiked with 50 µM HPG (final concentration of 20 nM), incubated in cultivation tubes with vent caps at *in situ* temperature in the dark for 4 h, fixed with 0.2 µm filtered 37% formaldehyde (2% final concentration) and stored at −80°C. Prior to the microscopic analysis, the samples were filtered and stained with DAPI (see Total Bacterial Abundance section). The filter slices were then processed according to click reaction protocols as follows: incubation in 200 µL of Click-It reaction buffer (154.5 µL Sigma water, 20 µL Click-It reaction buffer, 20 µL 10× reaction buffer additive, 4 µL copper (II) sulfate, 1.6 µL Alexa Fluor 488 azide) in the dark at room temperature for 30 min, followed by a Milli-Q water rinse and air-drying.

To quantify respiring or biomass-producing cells (RSG and HPG, respectively) in specific bacterial populations, additional filter slices from both RSG and HPG incorporation procedures were used for fluorescence *in situ* hybridization (FISH). The filter slices were labeled with taxa-specific oligonucleotide probes (Table S4) with Cy3 at the 5′-end (Biomers, Ulm/Donau, Germany) as described in reference (8).

All filter slices were analyzed using a Zeiss Axio Imager M2 epifluorescence microscope (Carl Zeiss AG, Oberkochen, Germany) at 1,250× magnification using the DAPI (Ex/Em = 358/461 nm) and the FITC (Ex/Em = 495/519 nm) filter sets as well as the Cy3

fluorophore (Ex/Em = 554/568 nm) filter in case of FISH. At least 20 fields were counted for each filter slice using the Automated Cell Measuring and Enumeration Tool (ACME-Tool2, M. Zeder, Technobiology GmbH, Buchrain, Switzerland). The total abundance of respiring or biomass-producing cells was determined as a simultaneous signal of DAPI and FITC channels. The taxa-specific abundance of respiring or biomass-producing cells was determined as a simultaneous signal of DAPI, FITC, and Cy3 channels.

## Isolation and sequencing of DNA

For the isolation of nucleic acids, a subsample of 0.5 L and 1 L was used from each Cteno-OM and control microcosm replicates, respectively, as well as 2L of the seawater inoculum. Bacterial biomass was collected from each sample using acid-washed, Milli-Q water-rinsed, and combusted filtration sets applying a low (<200 mbar) pressure. Total nucleic acids were extracted from the filters (36) as modified by reference (8). The extracted DNA was pooled from all Cteno-OM treatments and the control. Then, from all three DNA samples (Cteno-OM, control, and inoculum), a metagenomic DNA library was prepared using a Westburg kit with enzymatic shearing (Westburg Life Sciences, Utrecht, The Netherlands). Sequencing was performed on a single lane of the HiSeqV4 Illumina platform at the Vienna Biocenter Core Facilities (https://www.viennabiocenter.org/vbcf/next-generation-sequencing/).

## Metagenomic co-assembly and analysis

The metagenomes were investigated using Anvi'o v7.0 (37) using the default parameters for each step. The reads of all three metagenomic libraries were subject to quality filtering using Fastp v0.23.2 (38). The metagenomes were co-assembled using SPAdes v3.14.1 (39), followed by gene calling using PRODIGAL v2.6.3 (40). The identified genes were taxonomically classified using Kaiju v1.7.3 (41) against the NCBI RefSeq database from 26.02.2021 (42). Functional annotation of the genes was carried out using DIAMOND v2.0.15.153 (43) against the NCBI Clusters of Orthologous Groups of proteins (COG) database (44) and using KofamKOALA against the KEGG KOfam database (45).

The reads of each library were mapped to the co-assembled metagenome using BBmap v37.61 (sourceforge.net/projects/bbmap/), part of BBTools (46). Then, the metagenome was binned using CONCOCT v1.1.0 (47) and MetaBAT v2.16 (48) and merged using DAStool v1.1.6 (49). The resulting metagenome-assembled genomes (MAGs) were further refined (50). The completeness of metabolic KEGG modules (51, 52) in each MAG was estimated using the anvio program "anvi-estimate-metabolism" (see tutorial: https://merenlab.org/m/anvi-estimate-metabolism).

For determining the exact phylogeny of the selected MAG, complete genomes of *Pseudoalteromonas* were retrieved from the NCBI genome database. The phylogenetic tree was generated using trimAl v1.4.rev15 (53) and IQ-TREE v2.2.0.3 (54) based on single-copy genes identified using a hidden Markov model search in each genome.

The selected *P. phenolica* genomes (Table S5) were then annotated as described above and the pangenome was reconstructed using the Anvio program "anvi-pan-genome" (55) with MCL hierarchical clustering of the genes (56). The average nucleotide identity was calculated using PyANI v0.2.12 (57).

## Isolation and mass spectrometry analysis of proteins

For proteomic analysis, extraction of soluble proteins from ctenophore biomass and treatments' media (i.e., exo-metaproteomes) was performed as described in detail in Tinta et al. (8). For analyses of endo-metaproteomes, biomass was collected onto 0.22 µm polycarbonate filters. For the coastal endo-metaproteome, 3 L of the microbial inoculum was collected prior to the start of the experiment; 300 mL was collected from each ctenophore treatment and 1 L from each control treatment at the peak of the bacterial abundance. Protein extraction from collected cells was performed as follows: filters were ground into small pieces with a sterile metal spatula after submerging the tubes with

the filters into liquid nitrogen. Filter pieces were resuspended in lysis buffer (100 mM Tris-HCl pH 7.4, 1% SDS, 150 mM NaCl, 1 mM DTT, 10 mM EDTA) and cells were lysed with five freeze-and-thaw cycles. After centrifugation (20,000 × $g$ at 4°C for 25 min), the supernatant was transferred into a tube and proteins were co-precipitated with 0.015% deoxycholate and 7% trichloroacetic acid (TCA) on ice for 1 h and washed twice with ice-cold acetone. Dried protein pellets were resuspended with 50 mM TEAB buffer (Millipore Sigma, Burlington, MA, USA) and quantified using Pierce 660 nm Protein Assay Reagent (ThermoFisher Scientific, Rockford, IL, USA). Next, cysteines were reduced and alkylated with 10 mM DTT and 55 mM iodoacetamide (IAA), respectively.

For analyses of the exo-metaproteomes, the filtrate (<0.22 µm) of each sample was concentrated using a VivaFlow 200 (Sartorius, Göttingen, Germany) with 30 kDa and 5 kDa Molecular Weight Cut-Off (MWCO) to collect the high molecular weight (30 kDa–0.22 µm) and the low molecular weight (5 kDa–30 kDa) fraction, respectively. The high and low molecular-weight fractions were further concentrated to 250 µL using an Amicon (Millipore Sigma, Burlington, MA, USA) Ultra-15 Centrifugal Filter 30 kDa MWCO and 3 kDa MWCO Unit. Sample reducing agent NuPAGE (Invitrogen, Waltham, MA, USA) was added to the samples to reach 1× final concentration.

All samples (endo- and exo-metaproteomes) were re-precipitated using nine times the sample volume of 96% EtOH at −20°C overnight. Pellets were resuspended in 50 mM TEAB, followed by overnight in-solution trypsin (Roche, Basel, Switzerland) digestion (1:100, wt/wt) at 37°C. TFA was added to the samples at 1% final concentration to terminate trypsin digestion. Samples were desalted using Pierce C18 Tips (ThermoFisher Scientific, Rockford, IL, USA) according to the manufacturer's protocol. Prior to the LC-MS/MS analyses, digested peptides were dissolved in 0.1% formic acid and 2% acetonitrile and transferred into micro-inserts sealed with aluminum caps. Prior to the run, the concentration of peptides was measured using Pierce Quantitative fluorometric peptide assay (ThermoFisher Scientific, Rockford, IL, USA). The resulting peptides were sequenced on a Q-Exactive Hybrid Quadrupole-Orbitrap Mass Spectrometer (ThermoFisher Scientific, Rockford, IL, USA) at the Vienna Research Platform for Metabolomics & Proteomics and analyzed using the Proteome Discoverer v2.2.0.388 (ThermoFisher Scientific, Rockford, IL, USA) at the Life Science Computer Cluster (LiSC) of the University of Vienna (as previously described in reference (58). The tandem mass spectrometry spectra of proteins extracted from the Cteno-OM powder were searched using MASCOT v2.6.1 (59) against the *M. leidyi* transcriptome shotgun assembly project (NCBI TSA accession number GFAT01000000). The tandem mass spectrometry spectra of proteins extracted from the microcosms and the seawater inoculum were searched using SEQUEST-HT against the bacterial protein-coding genes of the co-assembled metagenome. Search parameters were as follows: enzyme—trypsin, fragment mass tolerance—0.8 Da, max. missed cleavages—2, fixed modifications—carbamidomethyl (Cys), optional modifications—oxidation (Met). Percolator parameters were as follows: max. delta Cn: 0.6, max. rank: 0, validation based on q-value, false discovery rate (calculated by automatic decoy searches) 0.05. Protein quantification was conducted using the chromatographic peak area-based label-free quantitative method.

## Statistical analyses

Statistical analyses were done in R v4.2.1 (R Core Team 2022) using RStudio v2022.02.3 (RStudio Team 2019). The metagenomic and metaproteomic data sets were combined and managed using "phyloseq" v1.40 (McMurdie and Holmes 2013). Dissimilarities between samples and statistical tests were carried out using "vegan" v2.6–2 (60). Protein enrichment analysis was performed using "DESeq2" v1.38.3 (61) on a variance-stabilized protein abundance matrix. To avoid overrepresentation, per protein only the COG assignment with the highest e-value was considered in the summary of the results. Mapping of the genes to KEGG pathways was carried out using "pathview" v1.38.0 (62).

## ACKNOWLEDGMENTS

We thank Sonja Tischler for performing mass spectrometry. We thank Barbara Mähnert for help with the experimental setup and the staff of Marine Biology Station Piran and the crew of RV Sagita for their help with sampling.

This project received funding from the European Union's Horizon 2020 Research and Innovation Program under the Marie Skłodowska-Curie Grant Agreement No. 793778. E.F. and G.J.H. were funded by the Austrian Science Fund (FWF) project I04978. T.T. was further supported by the Slovenian Research Agency under grant number ARRS J7-2599 and by the Slovenian Research Agency (Research Core Funding No. P1-0237).

E.F. performed the bioinformatic and statistical analyses and wrote the manuscript. T.T. and G.J.H. designed the experiments. T.T. and J.H.H. conducted the experiments and performed laboratory analyses. K.K. performed a chemical analysis of inorganic nutrients and dissolved organic matter. Z.Z. contributed and assisted with experimental design and preparation of samples for proteomic analysis. C.A. contributed and assisted with the preparation of samples for microscopy-based analysis. All authors contributed to the article and approved the submitted version.

## AUTHOR AFFILIATIONS

[1]Department of Functional and Evolutionary Ecology, Bio-Oceanography and Marine Biology Unit, University of Vienna, Vienna, Austria
[2]Marine Biology Station Piran, National Institute of Biology, Piran, Slovenia
[3]Department of Marine Microbiology and Biogeochemistry, NIOZ, Royal Netherlands Institute for Sea Research, Den Burg, the Netherlands
[4]Vienna Metabolomics & Proteomics Center, University of Vienna, Vienna, Austria

## AUTHOR ORCIDs

Eduard Fadeev  http://orcid.org/0000-0002-2289-2949
Zihao Zhao  http://orcid.org/0000-0001-7497-3276

## FUNDING

| Funder | Grant(s) | Author(s) |
| --- | --- | --- |
| EC \| European Research Council (ERC) | 793778 | Tinkara Tinta |
| Javna Agencija za Raziskovalno Dejavnost RS (ARRS) | J7-2599 | Tinkara Tinta |
| Austrian Science Fund (FWF) | I04978 | Gerhard J. Herndl |

## AUTHOR CONTRIBUTIONS

Eduard Fadeev, Data curation, Formal analysis, Investigation, Validation, Visualization, Writing – original draft, Writing – review and editing | Jennifer H. Hennenfeind, Formal analysis, Writing – review and editing | Chie Amano, Formal analysis, Writing – review and editing | Zihao Zhao, Conceptualization, Writing – review and editing | Katja Klun, Formal analysis, Writing – review and editing | Gerhard J. Herndl, Conceptualization, Funding acquisition, Project administration, Supervision, Writing – review and editing | Tinkara Tinta, Conceptualization, Funding acquisition, Investigation, Methodology, Project administration, Resources, Writing – original draft, Writing – review and editing

## DATA AVAILABILITY

The metagenomic raw sequences have been deposited in the European Nucleotide Archive (ENA) at EMBL-EBI under Project accession number PRJEB63998. The mass spectrometry proteomics data have been deposited to the ProteomeXchange Consortium (63) via the PRIDE (64) partner repository with the dataset identifier PXD043478.

Scripts for molecular data processing and statistical analyses can be accessed via Zenodo (https://www.zenodo.org/) under doi: 10.5281/zenodo.10453786.

## ADDITIONAL FILES

The following material is available online.

### Supplemental Material

**Supplemental material (mSystems01264-23-s0001.docx).** Supplemental figures and table legends.
**Table S1 (mSystems01264-23-s0002.xlsx).** Protein results table from Proteome Discoverer.
**Table S2 (mSystems01264-23-s0003.xlsx).** List of significantly enriched proteins and their annotations in Cteno-OM and Cotrol microcosms.
**Table S3 (mSystems01264-23-s0004.xlsx).** Unique and shared genes between the *P. phenolica* MAGs.
**Table S4 (mSystems01264-23-s0005.xlsx).** Taxa-specific oligonucleotide probes used for fluorescence *in situ* hybridization (FISH).
**Table S5 (mSystems01264-23-s0006.xlsx).** NCBI RefSeq accession numbers of *P. phenolica* used for the pangenome analysis.

### Open Peer Review

**PEER REVIEW HISTORY (review-history.pdf).** An accounting of the reviewer comments and feedback.

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
