## [Reviewer comments · mSystems]

Bacterial degradation of ctenophore *Mnemiopsis leidyi* organic matter

Eduard Fadeev, Jennifer Hennenfeind, Chie Amano-Sato, Zihao Zhao, Katja Klun, Gerhard Herndl, and Tinkara Tinta

Corresponding Author(s): Eduard Fadeev, Universitat Wien

Review Timeline:

Submission Date:

November 29, 2023

Accepted:

December 18, 2023

Editor: Michael Rappe

Reviewer(s): The reviewers have opted to remain anonymous.

Transaction Report:

DOI: <https://doi.org/10.1128/msystems.01264-23>

Re: mSystems01264-23 (Bacterial degradation of ctenophore *Mnemiopsis leidyi* organic matter)

Dear Dr. Eduard Fadeev:

Your manuscript has been accepted, and I am forwarding it to the ASM production staff for publication. Your paper will first be checked to make sure all elements meet the technical requirements. ASM staff will contact you if anything needs to be revised before copyediting and production can begin. Otherwise, you will be notified when your proofs are ready to be viewed.

Featured Image Submissions: If you would like to submit a potential Featured Image, please email a file and a short legend to mSystems@asmusa.org. Please note that we can only consider images that (i) the authors created or own and (ii) have not been previously published. By submitting, you agree that the image can be used under the same terms as the published article. File requirements: square dimensions (4" x 4"), 300 dpi resolution, RGB colorspace, TIF file format.

Sincerely,
Michael Rappe
Editor
mSystems

Reviewer #1 (Comments for the Author):

The authors have addressed the comments, especially including better presentation of proteomic data and stated the reason of focusing on dissolved rather than particulate fraction.

Only two minor suggestions:

Fig.4, either in legend or caption, briefly mention which colors are Gammaproteobacteria class and which colors are Alphaproteobacteria as these are mentioned in the result text.

Last paragraph in discussion highlighted the common bacterial lineage responding to two jellyfish-DOM, however, there were still difference in other parameters such as BB, BGE which may be related to the different biochemical properties of two OM, may mention this as well.

Reviewer #2 (Comments for the Author):

The authors have appropriately answered my previous comments, I have no further comments.